# Sevoflurane Does Not Promote the Colony-Forming Ability of Human Mesenchymal Glioblastoma Stem Cells In Vitro

**DOI:** 10.3390/medicina58111614

**Published:** 2022-11-08

**Authors:** Tomohiro Shoji, Mikio Hayashi, Chisato Sumi, Munenori Kusunoki, Takeo Uba, Yoshiyuki Matsuo, Kiichi Hirota

**Affiliations:** 1Department of Emergency Medicine, Kyoto Katsura Hospital, Kyoto 615-8256, Japan; 2Department of Human Stress Response Science, Institute of Biomedical Science, Kansai Medical University, Hirakata, Osaka 573-1010, Japan; 3Department of Physiology, Kansai Medical University, Hirakata, Osaka 573-1010, Japan; 4Department of Anesthesiology, Chibune General Hospital, Osaka 555-0034, Japan; 5Department of Anesthesiology, Kansai Medical University, Hirakata, Osaka 573-1010, Japan

**Keywords:** sevoflurane, glioblastoma stem cells, stemness

## Abstract

*Background and Objectives*: Clinically used concentrations of sevoflurane, an inhaled anesthetic, have been reported to significantly inhibit tumor growth. We investigated the effects of sevoflurane on sphere formation and the proliferation of human glioblastoma stem cells (GSCs) to determine whether sevoflurane exerts short- and long-term effects on human tumor cells. *Materials and Methods*: High-grade patient-derived GSCs (MD13 and Me83) were exposed to 2% sevoflurane. To evaluate the effect of sevoflurane on viability, proliferation, and stemness, we performed a caspase-3/7 essay, cell proliferation assay, and limiting dilution sphere formation assays. The expression of CD44, a cell surface marker of cancer stem-like cells in epithelial tumors, was evaluated using quantitative reverse transcription PCR. Differences between groups were evaluated with a one-way analysis of variance (ANOVA). *Results*: Sevoflurane exposure for 4 days did not significantly promote caspase 3/7 activity in MD13 and Me83, and cell proliferation was not observed after 5 days of exposure. Furthermore, prolonged exposure to sevoflurane for 6 days did not promote the sphere-forming and proliferative potential of MD13 and Me83 cells. These results suggest that sevoflurane does not promote either apoptosis, proliferative capacity, or the colony-forming ability of human mesenchymal glioblastoma stem cells in vitro. *Conclusions*: Sevoflurane at clinically used concentrations does not promote the colony-forming ability of human mesenchymal glioblastoma stem cells in vitro. It is very important for neurosurgeons and anesthesiologists to know that sevoflurane, a volatile anesthetic used in surgical anesthesia, would not exacerbate the disease course of GSCs.

## 1. Introduction

The impact of anesthetics on the prognosis of surgical cancer treatment has received considerable attention in recent years. Clinically used concentrations of sevoflurane, an inhaled anesthetic, have been reported to significantly inhibit tumor growth, migration, and invasion of cancer cells [1,2]. Sevoflurane inhibits cell migration and invasion by increasing miR-203 expression, which regulates the ERK/MMP-9 pathway in human colon cancer cells [1]. In human breast cancer cells, it significantly inhibits proliferation by arresting the cell cycle in the G1 phase or by increasing miR-203 expression to suppress cancer cell growth [2]. On the other hand, previous studies have shown that sevoflurane promoted the proliferation of cancers. Exposure to 2% sevoflurane at the clinically used concentration increased the proliferation of Lewis lung carcinoma cells by 9.2% compared to the control group [3]. However, exposure to sevoflurane did not affect the survival or tumor size in mice who were injected with Lewis lung carcinoma cells. In addition, sevoflurane promoted the proliferation of triple negative breast cancer cells in vitro via AKT, also known as protein kinase B [4]. Furthermore, sevoflurane promoted the growth of glioma stem cells through the activation of hypoxia-inducible factors [5]. The blockade of the PI3K/AKT pathways abrogated the growth-promoting effect. These conflicting results may be attributable to the heterogeneity of responses to the sevoflurane, depending on the cell type, concentration, or duration of exposure.

The prognosis of glioblastoma is poor, and surgery is the primary treatment modality. Accumulating evidence suggests that the stem-like properties of glioblastoma cells contribute to their therapeutic resistance and tumor recurrence [6]. The glioblastoma stem cells (GSCs) employ various molecular and cellular alterations to escape immune surveillance. Recent advances have classified GSCs into two subtypes called proneural and mesenchymal, which are generally consistent among the classification schemes [7]. Glioblastoma in the mesenchymal subclass are predominantly primary tumors that exhibit a worse prognosis compared to proneural tumors. The inducible T-cell co-stimulator ligand (ICOSLG), a member of the B7 family of ligands, was found in mesenchymal GSCs, which expressed CD44, a cell surface marker of cancer stem-like cells in epithelial tumors [6].

In this study, we investigated the effects of sevoflurane treatment on sphere formation and the proliferation of human mesenchymal GSCs to determine whether sevoflurane has short- and long-term effects on human tumor cells. Although the half-maximal inhibitory concentration of sevoflurane in GSCs has not been known, we used 2% sevoflurane in this study. This concentration corresponds to the minimum alveolar concentration (MAC) in clinical settings during anesthesia at the beginning of surgery.

## 2. Materials and Methods

### 2.1. Cell Culture and Sevoflurane Treatment

High-grade glioma patient-derived neurospheres (MD13 and Me83, which are known as human mesenchymal glioblastoma stem cells) were established from surgical specimens obtained by Dr. Nakano and his colleagues and molecularly characterized as previously described [6]. This study used cultured cells but did not use specific patient information. Glioblastoma stem cells (GSCs) were cultured in serum-free neurosphere medium [DMEM/F12 (048-29785; FUJIFILM Wako Pure Chemical, Osaka, Japan) supplemented with B27 (1:50, 130-097-263; Miltenyi Biotec, Bergisch Gladbach, Germany), penicillin/streptomycin (P4333; Sigma-Aldrich, St. Louis, MO, USA), L-glutamine (G7513, Sigma-Aldrich), basic fibroblast growth factor (20 ng/mL, 100-18B; PeproTech, Cranbury, NJ, USA), and epidermal growth factor (20 ng/mL; AF-100-15, PeproTech)] at 37 °C. Cells in the exponential growth phase were plated into 96-well ultralow plates (Corning, 3474) and treated with 2% sevoflurane (Maruishi Pharmaceutical, Osaka, Japan) in a sealed modular incubator chamber (MIC-101; Billups-Rothenburg, Del Mar, CA, USA). The cells in the sevoflurane groups were exposed to sevoflurane for 2, 4, and 6 h, and 6 days. Cells exposed to air were used as a control group.

### 2.2. Caspase-3/7 Assay

To evaluate the effect of sevoflurane on apoptosis, a caspase-3/7 assay was performed using Apo-ONE Homogeneous (Promega, Madison, WI, USA), according to the manufacturer’s protocol. The GSCs were plated in 96-well plates at a density of 2 × 10^3^ cells/well. The cells were treated with 2% sevoflurane and incubated at 37 °C for 4 days. After incubation, 50 µL of the reagent was added to each well, and the plate was incubated for 6 h. Absorbance was measured at 490 nm using a microplate reader (Bio-Rad, Hercules, CA, USA).

### 2.3. Cell Proliferation Assay

To evaluate the effect of sevoflurane on GSC proliferation, a cell proliferation assay was performed using Cell Count Reagent SF (Nacalai Tesque, Kyoto, Japan) according to the manufacturer’s protocol. The GSCs were plated in 96-well plates at a density of 2 × 10^3^ cells/well. The cells were treated with 2% sevoflurane and incubated at 37 °C for 3 or 5 days. After incubation, 5 µL of reagent was added to each well, and the cells were incubated for 4 h. The absorbance of each well was measured at 450 and 600 nm using a microplate reader (Bio-Rad).

### 2.4. In Vitro Limiting Dilution Sphere Formation Assay

Single GSCs were subjected to a limiting dilution from 32 to 0.25 cells per well in 96-well plates and cultured. Two protocols were tested. The group treated with 2% sevoflurane for 4 h (4 h) was compared with the control group (0 h). In addition, the group treated with 2% sevoflurane for 6 days (6 d) was compared with the control group (0 h). At 6 days after seeding, the number of wells without spheres was counted and plotted against the number of cells per well. The minimum number of cells required to form a sphere clump with a diameter of 50 μm was calculated using a logarithmic approximation graph. At the same time, the diameters of the formed sphere masses were measured using a microscope, and the areas were calculated. Extreme limiting dilution analysis was performed using software available at http://bioinf.wehi.edu.au/software/elda/ (accessed on 24 October 2014).

### 2.5. Quantitative Reverse Transcription PCR (qRT-PCR) Analysis

Total RNA was extracted using the RNeasy Mini Kit (Qiagen, Valencia, CA, USA) after treatment with 2% sevoflurane in sphere clumps following the three groups: sevoflurane 0 h + 37 ℃ 5% CO_2_ 8 h; sevoflurane 2 h + 37 °C 5% CO_2_ 6 h; and sevoflurane 6 h + 37 ℃ 5% CO_2_ 2 h. qRT-PCR (Roter-Gene) was performed for *CD44* (a cell surface marker of cancer stem-like cells in epithelial tumors; CD44 molecule), *ICOSLG* (inducible T cell costimulator ligand; ICOSLG), *PROM1* (CD133 antigen, also known as prominin 1; PROM1), *OLIG2* (oligodendrocyte transcription factor 2; OLIG2), and human 18S rRNA (QIAGEN QuantiTect Primer Assay: QT00199367). First-strand cDNA synthesis and real-time PCR were performed as previously described [8,9]. The PCR primer sequences are listed in Table 1.

### 2.6. Statistical Analysis

Data are presented as the means ± SD. Differences between groups were evaluated using a one-way analysis of variance. Statistical analyses were performed using the Prism software (GraphPad Software, Inc., La Jolla, CA, USA). Statistical significance was defined as *p* < 0.05.

## 3. Results

### 3.1. Sevoflurane Did Not Promote Caspase 3/7 Activity in GSCs

To evaluate the effect of sevoflurane on apoptosis in mesenchymal GSCs (MD13 and Me83), we performed a caspase-3/7 assay (Figure 1). The caspase 3/7 activity tended to increase in MD13, but decreased significantly in Me83 cells with sevoflurane treatment for 4 days. These results indicate that sevoflurane does not promote apoptosis in GSCs within 4 days.

### 3.2. Sevoflurane Did Not Promote the Proliferative Potential of GSCs

Sevoflurane treatment did not enhance the proliferation of MD13 and Me83 cells after five days (Figure 2A,B). These results indicate that sevoflurane does not promote the proliferative potential of mesenchymal GSCs.

### 3.3. Sevoflurane Treatment Did Not Promote the Colony-Forming Ability of GSCs

In both strains, there was no difference in the size of the spheres after sevoflurane treatment for 4 h and 0 h. Visually, the sizes of both MD13 and Me83 appeared smaller at 6 d than at 0 h, but there was no difference in the measured area (Figure 3A,B). There was no difference in the minimum number of cells required to form a spherical mass after sevoflurane treatment for 4 h (sevoflurane 4 h) compared with the control (sevoflurane 0 h) in MD13. The sevoflurane 4 h/sevoflurane 0 h slope ratio was 0.790 (Figure 3C). Similarly, the minimum number of cells was not different in Me83. The sevoflurane 4 h/sevoflurane 0 h slope ratio was 1.144 (Figure 3D). However, the minimum number of cells required to form a spherical mass tended to increase after 6 days of treatment with sevoflurane (sevoflurane 6 d) compared with the control (sevoflurane 0 h). The sevoflurane 6 d/sevoflurane 0 h slope ratios were 0.196 in MD13 and 0.585 in Me83 (Figure 3E,F). These results indicate that sevoflurane treatment does not promote the colony-forming ability of MD13 and Me83 cells.

### 3.4. Sevoflurane Treatment Did Not Promote the Proliferative Potential of Sphere

In MD13, the areas were 0.020 ± 0.010 mm^2^ (0 h) versus 0.025 ± 0.012 mm^2^ (4 h), and 0.008 ± 0.008 mm^2^ (0 h) versus 0.014 ± 0.013 mm^2^ (6 d). The sphere mass area was not different between the 4 h or 6 d treatment of MD13 (Figure 4A). Similar results were shown in Me83 (Figure 4B). The areas were 0.043 ± 0.018 mm^2^ (0 h) versus 0.037 ± 0.018 mm^2^ (4 h), and 0.028 ± 0.019 mm^2^ (0 h) versus 0.019 ± 0.011 mm^2^ (6 d). These results indicate that sevoflurane treatment does not promote the proliferative potential of MD13 and Me83 cells.

### 3.5. Sevoflurane Treatment Increased the Expression of CD44

The expression of *CD44* (CD44 molecule) was assessed using qRT-PCR. Sevoflurane treatment significantly increased the expression of *CD44* in MD13 and Me83 cells (Figure 5A,B). The expression of *ICOSLG* (inducible T cell costimulator ligand) increased after sevoflurane treatment in MD13 cells, but not in Me83 cells (Figure 5C,D). The expression of *PROM1* (prominin 1, CD133) and *OLIG2* (oligodendrocyte transcription factor 2) was not detected. The primer sequences of the four genes used are shown in Table 1.

## 4. Discussion

It has been widely indicated that anesthetic agents may affect the morphology, proliferation, and migration of malignant glioma cells [10,11]. Sevoflurane is frequently used in clinical anesthesia because of its rapid induction, rapid recovery, and reduced airway irritation [12]. Sevoflurane has been shown to inhibit cell proliferation, invasion, and migration in a variety of tumors, including lung cancer [13,14], breast cancer [2], and glioma [15]. It has also been reported that sevoflurane inhibits glioma cell viability, proliferation, and invasion dose-dependently in nude mice and may exert anticancer effects by inhibiting stemness, mitochondrial function, and tumor growth by activating the Ca^2+^/CaMKII/JNK cascade [16]. However, the effects of sevoflurane on glioma stem cells and their long-term prognosis in humans remain controversial.

A previous study demonstrated that sevoflurane promotes the expansion of human primary GSCs through hypoxia-inducible factors [5]. Sevoflurane exposure increased the expression of the CD133 total protein level in the human primary GSCs. However, we showed that sevoflurane did not promote the colony-forming ability, viability, and proliferation in human mesenchymal GSCs. The mesenchymal subclass is a CD133-negative subpopulation and exhibits the wild type of isocitrate dehydrogenase 1 (*IDH1*) [7]. Indeed, the expression of *PROM1* encoding CD133 was not detected in MD13 and Me83 by PCR analysis.

There is a report summarizing the effects of anesthetic in human cancer treatment [17]. Sevoflurane has been reported to exacerbate ovarian cancer [18] and promote lung metastasis [19], while others have reported that it suppresses glioma [20]. In U87MG glioma cells, sevoflurane at 2.5% attenuated the migratory ability of these cells, probably by downregulating the activity of matrix metalloproteinase-2 [20]. In this study, we demonstrated that sevoflurane does not promote the colony-forming ability of human mesenchymal glioblastoma stem cells, even after long-term exposure (Figure 3). This result strongly suggests that sevoflurane is not associated with tumor-grade progression, even if the amount of sevoflurane administered during multiple surgeries continues to accumulate in *IDH1* wild-type glioblastoma.

We do not elucidate why 2% sevoflurane exposure does not promote but rather suppresses the proliferative potential of stem cells in vitro, in contrast to the results in Lewis Lung Carcinoma [3]. Since we used glioblastoma stem cells exposed to sevoflurane for as long as 5 days, it is not surprising that the experimental model was different (Figure 2). There have been some recent reports of inhibition of cancer stemness in vitro. In mesenchymal stem-like cells, the inhibition of Integrin Alpha-6 (ITGA6) reduced radio resistance [21]. In pancreatic cancer stem cells, the inhibition of telomerase suppressed cancer stem cell markers and the colony-forming ability [22]. IFN-β induced apoptosis on GSCs [23]. It would also be interesting to study whether the 6-day continuous long-term exposure to 2% sevoflurane used in this experiment contributed to the inhibition of ITGA6 or telomerase or activation of IFN-β.

*CD44*, a marker of human GSCs, increased, indicating changes in mRNA expression in human glioma stem cells after several hours of exposure to sevoflurane (Figure 5). However, the less than 2-fold increase in CD44 expression detected in both cell lines may not have physiological significance when determined by qRT-PCR. The upregulation of *CD44* is insufficient to definitively conclude that sevoflurane does not affect the stemness of human glioma stem cells. The upregulation of *CD44* with sevoflurane treatment for 6 h was not consistent with the finding that sevoflurane did not promote the colony-forming ability of GSCs (Figure 3). The discrepancy implies that other molecules are involved in the stemness of GSCs and deserves dedicated future studies. We also found that the expression of *ICOSLG* increased after sevoflurane treatment in MD13 cells, but not in Me83 cells. The increase in *ICOSLG* expression might not contribute to the stemness of human GSCs. Despite sevoflurane treatment, the expression of *PROM1* and *OLIG2* was not increased, indicating that these genes are not important for mesenchymal MD13 and Me83 cells.

This study encountered some limitations. The difference in the minimum number of cells required to form a control sphere mass, shown in Figure 3C,E, is due to the variation in the number of seeded cells within the range of 32 to 0.25 cells, even among the same practitioners. In addition, future studies are needed to evaluate the effect of sevoflurane on glioma model animals.

## 5. Conclusions

Sevoflurane at clinically used concentrations does not promote the colony-forming ability of human mesenchymal glioblastoma stem cells in vitro. Additionally, sevoflurane increased CD44 expression less than 2-fold at the mRNA level, suggesting no physiological significance for GSCs. The prognosis of human glioblastoma is poor, and surgery is the primary treatment modality. It is very important for neurosurgeons and anesthesiologists to know that sevoflurane, a volatile anesthetic used in surgical anesthesia, would not exacerbate the disease course of GSCs. A retrospective study found no difference between intravenous and inhalation anesthesia on the overall survival of *IDH1* wild-type glioblastoma patients [24]. However, sevoflurane increased the risk of death in high-grade glioma patients with a Karnofsky performance status scale of less than 80 compared with propofol [25]. Further experiments on the effects of GSCs in vivo and in vitro are expected to be conducted in the future.

## Figures and Tables

**Figure 1 medicina-58-01614-f001:**
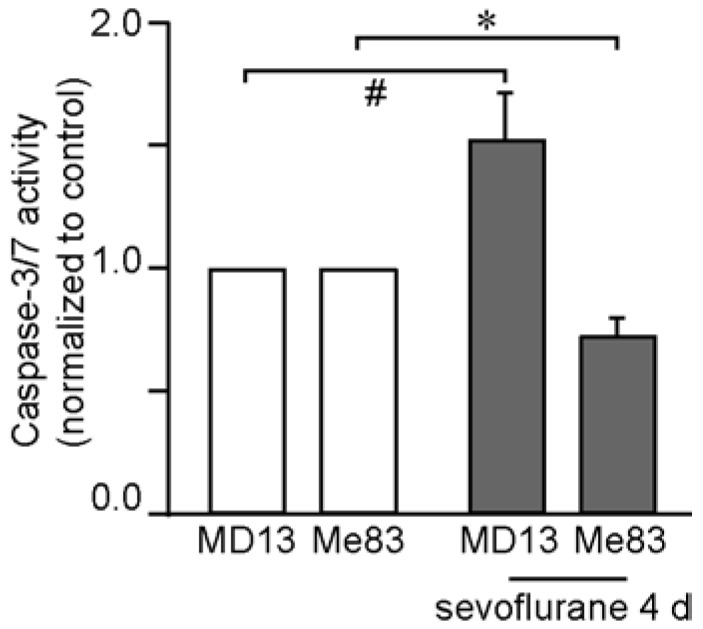
A caspase-3/7 assay was performed using Apo-ONE Homogeneous. The absorbance measured after 4 days of sevoflurane treatment (sevoflurane 4 d) was compared to the control (*n* = 6). Differences between groups were evaluated using one-way analysis of variance. Data are presented as mean ± SD. #: *p* = 0.05, *: *p* = 0.007.

**Figure 2 medicina-58-01614-f002:**
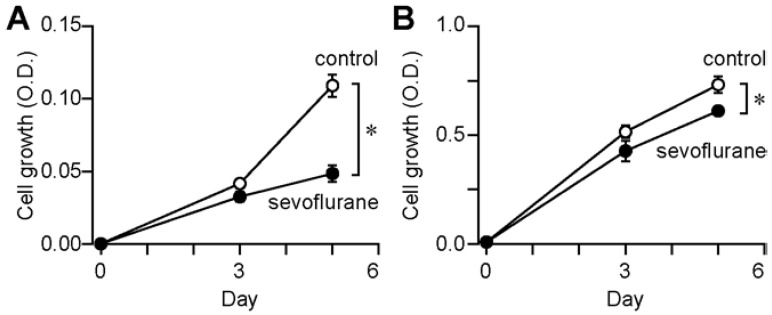
Cell proliferation assay. Growth of (**A**) MD13 and (**B**) Me83 cells. Data are presented as mean ± SD (*n* = 6). Differences between groups were evaluated using one-way analysis of variance. * *p* < 0.05, as compared with the value at 5 days.

**Figure 3 medicina-58-01614-f003:**
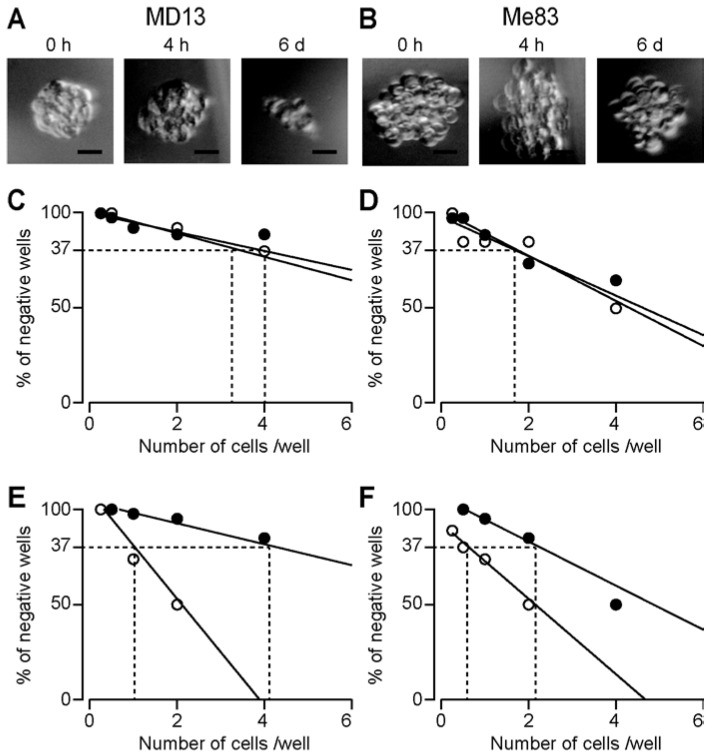
Limiting dilution sphere formation assay. Bars represent 50 μm. (**A**,**B**) Sphere clumps at 6 d after treatment with 2% sevoflurane for 0 h, 4 h, or 6 d. (**C**) Open and filled circles represent MD13 cells treated with 2% sevoflurane for 0 and 4 h, respectively. (**D**) Open and filled circles represent Me83 treatment with 2% sevoflurane for 0 and 4 h, respectively. (**E**) Open and filled circles represent MD13 treatment with 2% sevoflurane for 0 h and 6 days, respectively. (**F**) Open and filled circles represent Me83 treatment with 2% sevoflurane for 0 h and 6 days, respectively. The number of wells without spheres was counted and plotted against the number of cells per well.

**Figure 4 medicina-58-01614-f004:**
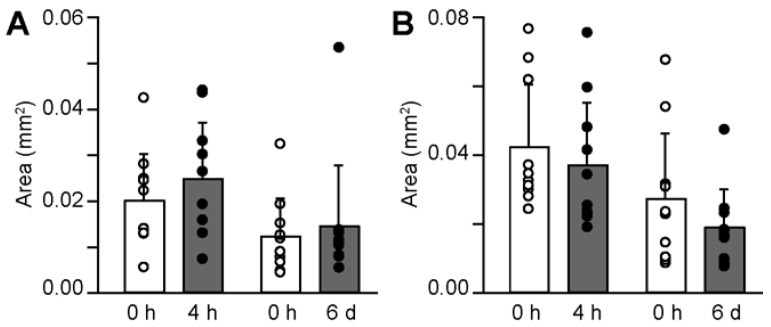
Sphere area. (**A**) Area of sphere clumps of MD13 cells at 6 days after treatment with 2% sevoflurane for 0 h, 4 h, or 6 d. (**B**) Area of sphere clumps of Me83 cells at 6 days after treatment with 2% sevoflurane for 0 h, 4 h, 6 d. Data are presented as mean ± SD (*n* = 10). Differences between groups were evaluated using one-way analysis of variance. All *p*-values for 4 h or 6 d compared to 0 h were greater than 0.05.

**Figure 5 medicina-58-01614-f005:**
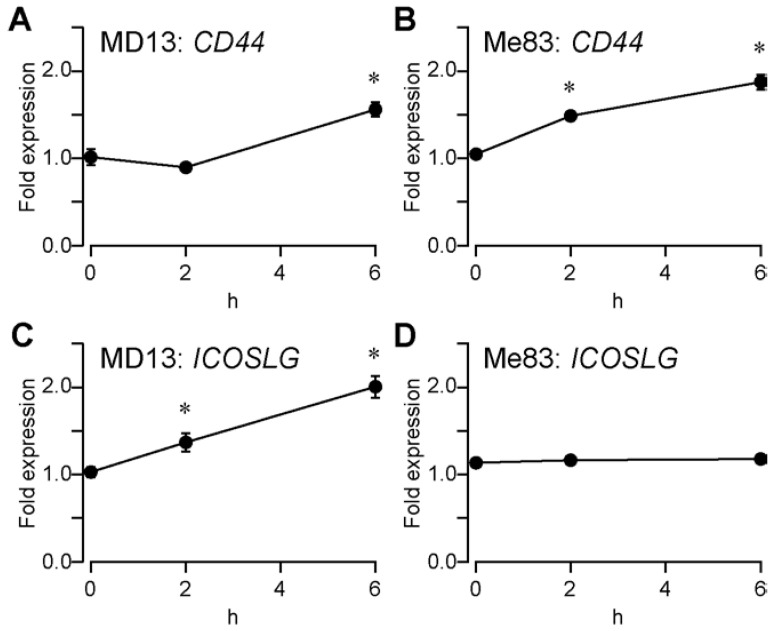
qRT-PCR analysis. GSCs were treated with 2% sevoflurane for 2 or 6 h. Then, the mRNA levels of *CD44*, *ICOSLG*, *PROM1*, *OLIG2*, and human 18S rRNA were assessed by *q*RT-PCR. Fold expression of *CD44* in (**A**) MD13 and (**B**) Me83 cells. Fold expression of *ICOSLG* in (**C**) MD13 and (**D**) Me83 cells. Their expression levels were normalized with human 18S rRNA. The expressions of *PROM1* and *OLIG2* were not detected. Data are presented as mean ± SD (*n* = 3). Differences between groups were evaluated using one-way analysis of variance. * *p* < 0.05, as compared with the value at 0 h.

**Table 1 medicina-58-01614-t001:** Primer sets used in qRT-PCR analysis.

Gene Name	Primer Sequence	Band Size (bp)
*CD44* (CD44)	FW: 5′-GCAAACACAACCTCTGGTCC-3′	129
	RV: 5′-CCCACACCTTCTTCGACTGT-3′	
*ICOSLG* (ICOSLG)	FW: 5′-GTCCTGGACTGCTCTTCCTG-3′	80
	RV: 5′-GCTGCCTACCATCGCTCTG-3′	
*PROM1* (CD133, PROM1)	FW: 5′-CATGCTCTCAGCTCTCCCGC-3′	102
	RV: 5′-TTTCTGTCTGAGGCTGGCTTG-3′	
*OLIG2* (OLIG2)	FW: 5′-TCGCATCCAGATTTTCGGGT-3′	142
	RV: 5′-CGGCAGAAAAAGGTCATCGG-3′	

## Data Availability

Not applicable.

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
