# Peer review of "Sevoflurane Does Not Promote the Colony-Forming Ability of Human Mesenchymal Glioblastoma Stem Cells In Vitro"

_medicina, 2022, doi:10.3390/medicina58111614_

Round 1
Reviewer 1 Report (Previous Reviewer 1)
After revision, the paper presents the current knowledge about describing the topic.
Reviewer 2 Report (Previous Reviewer 2)
The presented manuscript "Sevoflurane does not promote the colony-forming ability of human mesenchymal glioblastoma stem cells in vitro" describes the effects of sevoflurane on sphere formation and proliferation of human glioblastoma stem cells to determine whether sevoflurane exerts short- and long-term effects on human glioblastoma cells. The results suggest that sevoflurane does not promote apoptosis, proliferative capacity, and the colony-forming ability of human mesenchymal glioblastoma stem cells in vitro. Therefore, the authors concluded that sevoflurane at clinically used concentrations does not promote the colony-forming ability of human mesenchymal glioblastoma stem cells in vitro.
In the revised version, the significance and novelty of the study are well-substantiated. The Introduction and Discussion sections presented additional information on conflicting results demonstrating controversial responses to the sevoflurane depending on cell type, concentration, or duration of exposure.
The list of references was expanded. All methods are fully described. The Discussion section was improved. The Conclusions section was edited and expanded, making it more informative. The title of the manuscript was edited and now correctly reflects the results and conclusions.
Reviewer 3 Report (Previous Reviewer 3)
1. This manuscript is interesting and well-done.
2. Theme of this research, to suggest a better therapeutic approach of sevoflurane on human mesenchymal glioblastoma stem cells. In this study, author raised an interesting question treatment of sevoflurane and is well organized for readers to understand.
3. It's just my opinion, this manuscript was accepted in present form.
This manuscript is a resubmission of an earlier submission. The following is a list of the peer review reports and author responses from that submission.
Round 1
Reviewer 1 Report
However, the authors tried to improve their manuscript, the paper has been still looked like a draft. The corrections are not sufficient.
Reviewer 2 Report
The submitted manuscript has been edited. The necessary information has been added to the section Material and Methods. In the Results section, the necessary clarifications and information about the statistical analysis have been made. The results of the statistical analysis are also added to all Figures. Changes and clarifications have been made to the Introduction and Discussion sections to clarify some questions about the results.
Some minor comments:
- An increase in CD44 expression by no more than 2 times detected in both cell lines may not have physiological significance when determined by qRT-PCR. This explanation needs to be included in the Discussion on the Role of Increased CD44 Expression (Lines 219-230). This is also an additional argument for the conclusion that sevoflurane does not affect the stemness of human glioma stem cells.
- In the Abstract (L28) and Conclusions (L239), the sentence "Sevoflurane at clinically used concentrations dose not promote stemness in GSCs" needs typos correction (dose => does).
Reviewer 3 Report
- It is an honor for me to review this paper.
- Unfortunately, describing of all part of the paper were described not enough.
- This paper must increase the number of references in section of introduction and discussion.